# Antimicrobial and Anti-Biofilm Activity of Polymyxin E Alone and in Combination with Probiotic Strains of *Bacillus subtilis* KATMIRA1933 and *Bacillus amyloliquefaciens* B-1895 against Clinical Isolates of Selected *Acinetobacter* spp.: A Preliminary Study

**DOI:** 10.3390/pathogens10121574

**Published:** 2021-12-02

**Authors:** Munaf AL-Dulaimi, Ammar Algburi, Alyaa Abdelhameed, Maria S. Mazanko, Dmitry V. Rudoy, Alexey M. Ermakov, Michael L. Chikindas

**Affiliations:** 1Educational Laboratories, Baqubah General Hospital, Baqubah 32001, Iraq; munafthamer2233@gmail.com; 2Scholarship and Cultural Relations Department, University of Diyala, Baqubah 32001, Iraq; alyaa.maen@uodiyala.edu.iq; 3Biotechnology Department, College of Science, University of Diyala, Baqubah 32001, Iraq; 4Center for Agrobiotechnology, Don State Technical University, 344002 Rostov-on-Don, Russia; mary.bio@list.ru (M.S.M.); dmitriyrudoi@gmail.com (D.V.R.); amermakov@yandex.ru (A.M.E.); tchikind@sebs.rutgers.edu (M.L.C.); 5Health Promoting Naturals Laboratory, School of Environmental and Biological Sciences, Rutgers State University, New Brunswick, NJ 08904, USA; 6Department of General Hygiene, I.M. Sechenov First Moscow State Medical University, 119146 Moscow, Russia

**Keywords:** polymyxin E, CFS, spore-forming bacilli, antimicrobial combination, selected *Acinetobacter* spp. strain

## Abstract

*Acinetobacter* spp., the nosocomial pathogen, forms strong biofilms and is resistant to numerous antibiotics, causing persistent infections. This study investigates the antibacterial and anti-biofilm activity of polymyxin E alone and in combination with the cell-free supernatants (CFS) of the tested probiotic bacilli, *Bacillus subtilis* KATMIRA1933 and *Bacillus amyloliquefaciens* B-1895 against the selected *Acinetobacter* spp. starins. Three isolates of *Acinetobacter* spp., designated as *Acinetobacter* spp. isolate 1; *Acinetobacter* spp. isolate 2, and *Acinetobacter* spp. isolate 3, were collected from patients with burns, wounds, and blood infections, respectively. Bacterial identification and antibiotic susceptibility testing were conducted using the VITEK2 system. Auto-aggregation and coaggregation of the tested bacilli strains with the selected *Acinetobacter* spp. isolates were evaluated. A disk diffusion assay was used to identify the microorganism’s susceptibility to the selected antibiotics, alone and in combination with the CFS of the bacilli. The MIC and MBIC (minimum inhibitory and minimum biofilm inhibitory concentrations) of polymyxin E combined with bacilli CFS were determined. *Acinetobacter* spp. isolates were (i) sensitive to polymyxin E, (ii) able to form a strong biofilm, and (iii) resistant to the tested antibiotics and the CFS of tested bacilli. Significant inhibition of biofilm formation was noticed when CFS of the tested bacilli were combined with polymyxin E. The bacilli CFS showed synergy with polymyxin E against planktonic cells and biofilms of the isolated pathogens.

## 1. Introduction

*Acinetobacter* spp. is a Gram-negative, obligate aerobe, cocco-bacilli, and one of the most prevalent causative agents of several hospital and community-acquired infections [1]. This bacterium is related to skin, soft tissue, and urinary tract infections, in addition to meningitis, bacteremia, and pneumonia [2]. *Acinetobacter* spp. is one of the multidrug-resistant (MDR) ESKAPE pathogens, which include *Escherichia coli*, *Staphylococcus aureus*, *Klebseilla pneumoniae*, *Acinetobacter baumannii*, *Pseudomonas aeruginosa,* and *Enterococcus faecalis* [3].

*Acinetobacter* spp. cells have three main mechanisms of antibiotics resistance: they (i) produce antibiotic-hydrolyzing enzymes, (ii) interrupt binding of antibiotics to the target site of a bacterial cell, and (iii) alternate their target site or modify their cellular functions to avoid an antibiotics’ activity [4]. Furthermore, biofilm formation by *Acinetobacter* spp. is the most important virulence factor, playing an important role in bacterial survival, infection, and antibiotic resistance [3,5].

The “old generation” of antibiotics, such as polymyxins, are commonly used as the drugs of choice to eliminate *Acinetobacter* infections [6]. Polymyxin E (or colistin) has recently been used as a “last line” therapeutic substance to control the growth of multidrug resistant Gram-negative bacteria [7]. Polymyxin E, a cationic antimicrobial peptide, interacts with the lipid A moiety of bacterial lipopolysaccharides (LPS) and eventually disrupts the outer membrane of bacterial pathogens [8]. However, monotherapy has been reported as a less effective protocol compared to antimicrobial combinations, which are recommended by the National Institutes of Health (NIH) as a potential integrative therapeutic option [9]. Therefore, effective approaches using antimicrobial combinations are urgently required as alternative and safe strategies to combat bacterial resistance to antibiotics. One of the suggested methods is using probiotics and their metabolites as antimicrobial substances in combination with conventional antibiotics to increase the sensitivity of pathogenic strains [10].

Probiotics are live microorganisms that provide health benefits when administered in appropriate amounts [11]. Probiotics play a key role in the regulation of the host immune system by stimulating cytokine production and cellular activity and inhibiting the clustering of pathogens [12]. In addition, the therapeutic properties of probiotics can be attributed to the production of a variety of antibacterial agents, such as short-chain fatty acids, organic acids, ethanol, hydrogen peroxide, and bacteriocins [13].

The combination of polymyxin E with other antibiotics, which is widely used by physicians in critical patients, is reported to have a synergistic effect [6]. These combinations have several advantages, such as (i) using lower concentrations of antimicrobials with greater activity, (ii) reducing their cost, and (iii) limiting toxic side effects (nephrotoxicity and neurotoxicity) [10]. This study aimed to evaluate, in vitro, the antibacterial and anti-biofilm activities of the tested probiotic bacilli and their cell-free supernatants (CFS) alone and in combination with polymyxin E against selected *Acinetobacter* spp. strains. In addition, we demonstrated the coaggregation potential of probiotic strains with the *Acinetobacter* spp. isolates.

## 2. Results

### 2.1. Bacterial Isolation and Identification

This study identified three isolates of the selected *Acinetobacter* spp. in samples taken from hospitalized patients with blood, burn, or wound infections. The clinical isolates grown on blood agar were small, smooth, opaque, raised, creamy, non-hemolytic colonies. On MacConkey agar, the bacterial colonies were pale pinkish, with small size and regular edges. All isolates were grown at 37 and 44 °C, as required. The tested isolates were positive for catalase production and the Simmons citrate test but variable regarding urease production. They were negative to oxidase, Voges–Proskauer, methyl red, and indole production tests. Due to their inability to ferment sugars, their growth on Triple Sugar–Iron (TSI) agar was alkaline with no gases production. The isolates showed no lactose fermentation on the MacConkey agar plate.

### 2.2. Bacterial Identification and Antibiotic Susceptibility Using VITEK 2 System

Bacterial identification was confirmed by using a Gram-negative Identification (GN-ID) card. The sensitivity of the VITEK 2 system showed that clinical isolates were 97–99% *Acinetobacter* spp. which was in agreement with the phenotypic and initial biochemical characteristics described earlier. In addition, to confirm bacterial identification, the VITEK 2 system was used to evaluate the antibiotic susceptibility of the selected *Acinetobacter* spp. strains using an antibiotics susceptibility Gram-negative (AST-GN) card. In this method, 18 antimicrobial agents were evaluated against the *Acinetobacter* spp. isolates. We reported that the isolated pathogens were highly tolerant to most of the tested antibiotics. The three isolates were tolerant to 11 tested antibiotics, while two *Acinetobacter* spp. isolates 1 and 2 were tolerant to trimethoprim-sulfamethoxazole, gentamicin, tobramycin, imipenem, and amikacin. *Acinetobacter* spp. isolates 2 and 3 were sensitive to tigecycline and minocycline, while all the isolates were sensitive to polymyxin E. Our findings showed that the three isolates were susceptible to polymyxin E. Only *Acinetobacter* spp. isolate 1 was resistant to tigecycline and minocycline. In regard to trimethoprim/sulfamethoxazole, we noticed that only *Acinetobacter* spp. isolate 3 was inhibited. Fluoroquinolones resistance was reported in the three selected isolates of *Acinetobacter* spp. Our findings showed that *Acinetobacter* spp. isolates 1 and 2 were resistant to amikacin, gentamicin, and tobramycin (Table 1).

### 2.3. Probiotic Strains Were Tolerant to Polymyxin E

In this study, six antibiotics were evaluated against the tested bacilli strains using the disc diffusion method (Kirby–Bauer). Our findings showed that bacilli strains were susceptible to the majority of the selected antibiotics (Table 2), while they were tolerant to polymyxin E. In addition, *B. amyloliquefaciens* B-1895 showed tolerance to amikacin. Based on the above-mentioned data, polymyxin E was selected in this study to be used alone and in combination with the probiotics CFS to identify the nature of antimicrobial interactions against the selected *Acinetobacter* spp. strains.

### 2.4. Minimum Inhibitory Concentrations (MICs)

The MICs of polymyxin E and the tested probiotic CSF were determined for the selected *Acinetobacter* spp. isolates 1, 2, and 3 using the broth micro-dilution method. The MICs of polymyxin E were 3.13 μg/mL for *Acinetobacter* spp. isolate 3, whereas it was 6.25 μg/mL for both the *Acinetobacter* spp. isolates 1 and 3 (Figure 1). At these concentrations, bacterial inhibition growth was significantly inhibited (*p* < 0.001).

No MIC values were determined for the CFS of *B. subtilis* KATMIRA1933 against the three isolated pathogens, even when the highest concentration (50%) was used. Compared to the control (bacterial growth without treatment), only *Acinetobacter* spp. isolate 2 growth was inhibited (*p* < 0.05) at the concentration of 25%. The *Acinetobacter* spp. isolates 2 and 3 were significantly inhibited (*p* < 0.01) at a concentration of 50%, while the growth of *Acinetobacter* spp. isolate 1 was not influenced by the CFS of the tested *Bacillus* strains (Figure 2). Similarly, no MIC value for the CFS of *B. amyloliquefaciens* B-1895 was identified against the three isolates. However, significant growth inhibition (*p* < 0.01) of *Acinetobacter* spp. isolates 2 and 3 was noticed at a concentration of 12.5%. At the highest concentration of *B. amyloliquefaciens* B-1895 CFS (50%), the bacterial isolates growth was dramatically inhibited (*p* < 0.001) (Figure 3).

### 2.5. The Bacilli Strains Auto-Aggregated and Co-Aggregated with Isolated Pathogens

Kinetic measurements of auto-aggregation and coaggregation of the probiotic bacilli strains with the three selected *Acinetobacter* spp. isolates were determined at 0, 4, and 24 h timepoints using an automated microplate reader at a wavelength of 630 nm. After 4 h of incubation, the percentages of auto-aggregation of *B. amyloliquefaciens* B-1895 and *B. subtilis* KATMIRA1933 were 25.5% and 22.76%, respectively. We noticed that, after 24 h, the percentages of auto-aggregation increased as follows: *B. amyloliquefaciens* B-1895 and *B. subtilis* KATMIRA1933 were 95.7% and 82.4% (Table 3). In this study, the percentages of *Acinetobacter* spp. isolates 1, 2, and 3 auto-aggregation after 4 h of incubation were 24.5%, 20.58%, and 28.97%, respectively, whereas, after 24 h, they were lower, at 0.48%, 8.4%, and 30.5%, respectively (Table 3).

Regarding bacterial coaggregation, after 4 h, a high coaggregation percentage was observed when *B. subtilis* KATMIRA1933 was mixed with *Acinetobacter* spp. isolates 1, 2, and 3, at 33.43%, 31.89%, and 34%, respectively. When *B. amyloliquefaciens* B-1895 was mixed with *Acinetobacter* spp. isolates 1, 2, and 3, the coaggregation percentages were 23.98%, 29.39%, and 17.15%, respectively. After 24 h incubation, the percentages of coaggregation were higher compared to 4 h. *B. subtilis* KATMIRA1933 was co-aggregated with *Acinetobacter* spp. isolates 1, 2, and 3 at 60.1%, 53.16%, and 62.8%, respectively, and the coaggregation of *B. amyloliquefaciens* B-1895 with *Acinetobacter* spp. isolates 1, 2, and 3 was 50.57%, 55.64%, and 50.67%, respectively. The highest scores of coaggregation that appeared after 24 h were stained and photographed using light microscopy, as shown in Figure 4.

### 2.6. Minimum Biofilm Inhibitory Concentrations (MBIC)

MBICs were determined using the broth-microtiter dilution method. Regarding polymyxin E, there was biofilm formation inhibition at 1.65 and 3.13 µg/mL as compared to the positive control (Figure 5). The MBIC of polymyxin E was 3.13 µg/mL for the selected *Acinetobacter* spp. isolate 3 with 95–100% biofilm inhibition, while the MBIC value for *Acinetobacter* spp. isolates 1 and 2 was 6.25 µg/mL at which a significant reduction (*p* < 0.001) in biofilm formation was reported, with 85% and 83.4% inhibition for *Acinetobacter* spp. isolates 1 and 2, respectively (Figure 5).

In our study, no MBIC value for the *B. subtilis* KATMIRA1933 CFS was determined against the selected isolates, even when the highest concentration of 50% was used. A concentration of 50% CFS of *B. subtilis* KATMIRA1933 prevented 7.3%, 23.3%, and 22.5% of the biofilm formation by *Acinetobacter* spp. isolates 1, 2, and 3, respectively, with a significant difference (*p* < 0.01) (Figure 6). Similarly, no MBIC was determined when *B. amyloliquefaciens* B-1895 CFS was used. However, a slight reduction (5–10%) was reported in biofilm formation by *Acinetobacter* spp. isolate 1 at 12.5% and 25% (Figure 7). At 50%, a significant (*p* < 0.01) reduction (48.7%) was reported in biofilm formed by the *Acinetobacter* spp. isolate 1. In the same regard, 50% CFS of *B. amyloliquefaciens* B-1895 significantly (*p* < 0.05) reduced biofilm formation by 28.4% and 33.6% for *Acinetobacter* spp. isolates 2 and 3, respectively.

### 2.7. The Polymyxin E Synergizes with Bacilli CFS

Initially, a modified Kirby–Bauer method was used to determine the antibiotic susceptibility of the selected *Acinetobacter* spp. isolates to polymyxin E alone and combined with the CFS of tested probiotic strains.

We found a significant increase in the inhibition of the selected *Acinetobacter* spp.strains growth when the CFS of the two bacilli strains were combined with polymyxin E, compared to using the antibiotic alone (Figure 8). Polymyxin E alone produced a zone of inhibition around the *Acinetobacter* spp. isolates 1, 2, and 3 (11 mm for each), while *Acinetobacter* spp. isolate 3 had a 10 mm zone of inhibition around the same antibiotic disc (Figure 8). When the CFS of *B. subtilis* KATMIRA1933 was combined with polymyxin E, the diameters of the zones of inhibition for *Acinetobacter* spp. isolates 1, 2, and 3 were 12, 14, and 16 mm, respectively. In the same regard, the zone inhibition was significantly increased (13 mm; *p* < 0.01) when the polymyxin E disc was combined with the CFS of *B. amyloliquefaciens* B-1895.

### 2.8. Probiotic Strains Synergized with Polymyxin E against the Planktonic and Biofilm Cells of the Selected Acinetobacter spp. Isolates

A checkerboard assay was used to evaluate the antimicrobial combinations between polymyxin E and the CFS of the tested probiotic strains against the selected *Acinetobacter* spp. isolates. After 24–36 h incubation, the MICs and MBICs of antimicrobial combinations were determined using a microplate reader at OD 630 nm. Our results showed that the antimicrobial activity of polymyxin E was complimented with the CFS of tested bacilli against both planktonic and biofilm cells of selected *Acinetobacter* spp. isolates. We noticed that the MIC and MBIC values of antibiotics in combination with bacilli CFS were lower than using each of them alone.

Synergistic activity was identified when polymyxin E was combined with the CFS of the probiotic strains against the *Acinetobacter* spp. isolate 1. The MIC of polymyxin E was 2-fold lower, and the CFS of *B. subtilis* KATMIRA1933 was more than 17-fold lower than their MICs when used alone (3.13 μg/mL of polymyxin E in combination instead of 6.25 μg/mL alone and 1.6–3.13% of CFS in combination compared to more than 50% when used alone) (Figure 9(B1)). Similarly, the MIC of polymyxin E was 2-fold and the CFS of *B. amyloliquefaciens* B-1895 was more than 8–16-fold lower than using them alone (3.13 μg/mL of polymyxin E in combination instead of 6.25 μg/mL alone and 1.65–6.25% of CFS in combination compared to more than 50% when it was used alone) (Figure 9(A1)). The total ∑FIC (fractional inhibitory concentration) showed synergism between polymyxin E and the CFS of the probiotic strains against the *Acinetobacter* spp. isolate 1. The ∑FIC was 0.527, and 0.511 when polymyxin E was combined with *B. subtilis* KATMIRA1933 and *B. amyloliquefaciens* B-1895, respectively, against biofilm cells.

Regarding the *Acinetobacter* spp. isolate 2, synergistic activity was identified when polymyxin E was combined with the CFS of *B. subtilis* KATMIRA1933 but not with the CFS of *B. amyloliquefaciens* B-1895. The MIC of polymyxin E was 2-fold lower, and the CFS *B. subtilis* KATMIRA1933 was more than 17-fold lower than their individual MIC values (3.13 μg/mL of polymyxin E in combination instead of 6.25 μg/mL alone and 1.6–3.13% of CFS in combination compared to more than 50% when used alone) (Figure 9(B2)). An additive effect was reported when polymyxin E was combined with the CFS of *B. amyloliquefaciens* B-1895. The MIC of polymyxin E was not changed (6.25 μg/mL), while the MIC of the bacilli CFS was more than 16-fold lower (1.65–6.25% of CFS in combination compared to more than 50% when used alone) (Figure 9(A2)). The total ∑FIC showed synergism between polymyxin E and the CFS of *B. subtilis* KATMIRA1933 but not with the CFS of *B. amyloliquefaciens* B-1895 against the *Acinetobacter* spp. isolate 2. The ∑FIC was 0.516 and 1.02 when polymyxin E was combined with *B. subtilis* KATMIRA1933 and *B. amyloliquefaciens* B-1895, respectively, against biofilm cells.

Synergistic activity was also noticed when polymyxin E was combined with the CFS of the probiotic strains against the *Acinetobacter* spp. isolate 3. The MIC of polymyxin E was 4-fold lower and the CFS of *B. subtilis* KATMIRA1933 was more than 8–16 fold lower (1.63 μg/mL of polymyxin E in combination instead of 6.25 μg/mL alone and 1.6–3.13% of CFS in combination compared to more than 50% when it was used alone) (Figure 9(B3)). Similarly, the MIC of polymyxin E was 4-fold lower, and the CFS of *B. amyloliquefaciens* B-1895 was more than 8–16 fold lower when combined (1.63 μg/mL of polymyxin E in combination instead of 6.25 μg/mL alone and 3.13–12.5% of CFS in combination compared to more than 50% when used alone) (Figure 9(A3)). The total ∑FIC showed synergy between polymyxin E and the CFS of the probiotic strains against the *Acinetobacter* spp. isolate 3. The ∑FIC was 0.381 and 0.272 when polymyxin E was combined with *B. subtilis* KATMIRA1933 and *B. amyloliquefaciens* B-1895, respectively, against biofilm cells.

## 3. Discussion

Recently, there has been an increase in the number of multidrug-resistant *Acinetobacter* spp. isolates (about 90%) from burns and other wounds [14]. The high prevalence of multidrug-resistant *Acinetobacter* spp. infections is associated with several factors, such as the (a) acquisition of nosocomial pathogens during long-term hospitalization, (b) delayed administration of antimicrobial agents, and (c) patients having immunosuppressive factors [15]. On MacConkey agar, the bacterial colonies appeared as described by AL-Dahlaki [16]. Importantly, the ability to grow at 44 °C is a feature that distinguishes *Acinetobacter* spp. isolates from the rest of this genus [6]. The VITEK 2 system was used to confirm bacterial identification, and additionally, to evaluate the antibiotic susceptibility of the selected *Acinetobacter* spp. isolates. The VITEK 2 identification system is unable to discriminate between *A. baumannii* and *A. nosocomialis*, which requires an analysis of the *rpo*B gene. In this study, the isolated pathogens were indicated as “selected *Acinetobacter* spp. isolates”. The future mechanistic study of the newly isolated pathogens will include their taxonomic identification using the appropriate approach.

Our results demonstrated that the bacterial isolates were tolerant to the imipenem and meropenem β-lactam antibiotic classes (Table 1). AL-Dahlaki [16], found that 95% of the tested *Acinetobacter* spp. isolates were resistance to carbapenems, imipenem, and meropenem. Our findings showed that the tested pathogens were resistant to ampicillin and piperacillin/tazobactam (100%). These results were in agreement with a study of AL-Dahlaki [16], who found that 100% of *Acinetobacter* spp. isolates showed resistance to piperacillin/tazobactam. Furthermore, Raut et al. [17] and Pal et al. [18] found that *Acinetobacter* spp. isolates were mostly resistant to piperacillin/tazobactam and survived in the presence of ampicillin.

In regard to cephalosporins, our data were in agreement with Pal et al. [18], who found that 100% of *Acinetobacter* spp. isolates were resistant to cephalosporins.

β-lactamase production by *Acinetobacter* spp. isolates plays an important role in their tolerance to β-lactam antibiotics. This enzyme breaks down the amide bond of the β-lactam ring, causing inactivation of these types of antibiotics. Moreover, modifying penicillin-binding proteins in bacterial cells will decrease the permeability of the outer membrane porins and excretion of β-lactam antibiotic from the cell by the efflux pump [6].

Our findings showed that the three selected isolates were susceptible to polymyxin E. Only *Acinetobacter* spp. isolate 1 was resistant to tigecycline and minocycline. These findings were similar to those of Rahimi et al. [19], who reported that the isolated *Acinetobacter* spp. were sensitive to polymyxin E. In addition, 94% and 73% of isolates were sensitive to tigecycline and minocycline. The study of Raut et al. [17] showed that *A. baumannii* isolates were 100% susceptible to polymyxin E and tigecycline.

Polymyxin E, a bactericidal substance, is widely used to control MDR isolates by disrupting bacterial cell membranes [17]. It has a positively charged cationic region that binds to the hydrophilic portion of bacterial lipopolysaccharides leading to the eventual loss of cellular membrane integrity [20]. Tetracyclines and glycylcyclines inhibit protein synthesis of bacterial cells by preventing aminoacyl-tRNA binding to the ribosome [21]. In the same regard, *A.*
*baumannii* possesses *TetA* and *TetB* genes, which control the efflux of antibiotics outside the bacterial cell [22].

In regard to trimethoprim/sulfamethoxazole, we noticed that only *Acinetobacter* spp. isolate 3 was inhibited (Table 1). The presence of dihydrofolate reductases (DHFR and FolA) in *Acinetobacter* spp. isolates plays an important role in bacterial resistance to trimethoprim [23]. Fluoroquinolone resistance was reported in the three selected isolates of *Acinetobacter* spp. Fluoroquinolones are broad-spectrum antibiotics; however, an increase in bacterial resistance to fluoroquinolones has been reported over the past several years [17]. A major mechanism for quinolone resistance was identified in the mutated genes *gyrA* and *parC*, which led to phenotypic changes in DNA gyrase and topoisomerase IV, reducing antibiotic affinity [24].

Our findings show that *Acinetobacter* spp. isolate 1 and *Acinetobacter* spp. isolate 2 were resistant to amikacin, gentamicin, and tobramycin (Table 1). These data are in agreement with the work of AL-Dahlaki [16], who reported that most *A. baumannii* isolates were resistant to amikacin, gentamicin, and tobramycin.

Aminoglycoside resistance could be related to aminoglycoside-modifying enzymes (AMEs) produced by *Acinetobacter* spp. These enzymes change the corresponding functional groups of aminoglycosides and disrupt the binding capacity of these antibiotics at their ribosomal target sites [25]. Moreover, aminoglycosides resistance is associated with 16S rRNA methylase genes, which alter the bacterial-binding site of aminoglycosides within the 30S ribosomal subunit. Methylases stimulate high-level resistance to aminoglycosides, including amikacin, gentamicin, and tobramycin [25].

In regard to the susceptibility of the probiotic strains to antibiotics, six antibiotics were evaluated using the disc diffusion method (Kirby–Bauer). Our findings showed that bacilli strains were susceptible to the majority of the selected antibiotics (Table 2), while they were tolerant to polymyxin E. In addition, *B. amyloliquefaciens* B-1895 showed tolerance to amikacin. Polymyxin E is a non-ribosomal peptide produced by *Bacillus polymyxa*, a soil bacterium, as a secondary metabolite with bactericidal activity against Gram-negative bacteria [7].

According to the Clinical and Laboratory Standards Institute (CLSI) [26] guidelines, the selected *Acinetobacter* spp. isolates were classified into two major classes; MDR (resistant to ≥3 of all antibiotic categories) or XDR (resistant to all antibiotics except two or fewer belong to the same category). *Acinetobacter* spp. isolate 3 was MDR, while both *Acinetobacter* spp. 1 and 2 were XDR isolates. A complete picture of antibiotic resistance for the tested isolates was created according to the criteria outlined by Magiorakos et al. [27]. Our results agree with the data of Rahimi et al. [19], who found that 76% of *Acinetobacter* spp. isolates exhibited an XDR phenotype. Recently, several studies highlighted the obstacle of increasing resistance to antibiotics and the global spread of multidrug-resistant bacteria [28]. An increasing trend in the emergence of XDR strains was also reported over the last decade in Iran [29]. The development of antibiotic resistance in *Acinetobacter* spp. strains is related to the ability to form biofilms. In the current study, we found that the three isolated *Acinetobacter* spp. were capable of forming strong biofilms. The ability of *Acinetobacter* spp. to form biofilms could explain the outstanding resistance to antibiotics, long-time survival in harsh environments, and tolerance for disinfectants and/or desiccation on abiotic surfaces [30]. Several factors are involved in biofilm formation and antibiotic resistance of *Acinetobacter* spp., including biofilm-associated proteins (Bap), efflux systems (AdeABC, AdeFGH, and AdeIJK), quorum sensing systems, and motility by pili [31].

The broth micro-dilution method was used to determine the MIC of polymyxin E and the tested probiotic CFS against *Acinetobacter* spp. isolates 1, 2, and 3. The data showed that *Acinetobacter* spp. isolate 3 was more sensitive to polymyxin E compared to *Acinetobacter* spp. isolates 1 and 2, the MICs were 3.13–6.25 μg/mL (Figure 1). Our results were similar to a study by Sato et al. [32], who reported that the MIC of polymyxin E against *A. baumannii* was 4 μg/mL. However, a study by Lin et al. [33] found a lower MIC value for polymyxin E (1 μg/mL) against *A. baumannii*. The variation in the MIC values might be related to the source and manufacture of the antibiotic, in addition to differences in experimental designs and conditions.

Regarding the MIC for the CFS of *B. subtilis* KATMIRA 1933, only the *Acinetobacter* spp. isolate 2 showed growth inhibition (*p* < 0.05) at a concentration of 25%. The *Acinetobacter* spp. isolates 2 and 3 were inhibited significantly (*p* < 0.01) at a concentration of 50% (Figure 2). Significant growth inhibition (*p* < 0.01) of *Acinetobacter* spp. isolates 2 and 3 was found at 12.5%, while at the highest concentration of *B. amyloliquefaciens* B-1895 CFS (50%), the bacterial isolates were dramatically inhibited (*p* < 0.001) (Figure 3). Efremenkova et al. [34] proposed that *Bacillus* strain 534 can produce active substances with different molecular sizes that target the cellular envelope of the pathogenic microorganisms. Several studies have been performed to evaluate the antimicrobial activity of the metabolites extracted from various species of probiotics against *A. baumannii*. Shin and Eom [3] referred to the antimicrobial activity of *C. butyricum* CFS against *A. baumannii* strains. In the presence of 50% of *C. butyricum* CFS, 98.51% of *A. baumannii* growth was inhibited. A study by Soltan et al. [35] showed that lactobacilli prevented the growth of *A. baumannii* and *P. aeruginosa*. During the stationary phase of growth, lactobacilli and bacilli strains secrete weak organic acids, bacteriocins, and biosurfactants. Bacteriocins are antimicrobial peptides ribosomally produced by virtually all microorganisms [36]. These peptides interact with the bacterial cell surface and cell membrane, leading to cell permeabilization and pore formation, and eventually, depletion of intracellular ATP (because of the collapse of the proton motive force) and cellular death (after leakage of intracellular substrates-please, see the review of Kumariya et al. [37]).

For coaggregation, the results illustrated that the percentages of coaggregation after 24 h incubation were higher compared to 4 h. Similarly, Algburi et al. [38] reported high levels of coaggregation of pathogenic *P. mirabilis,* isolated from urinary tract infections, with *B. amyloliquefaciens* B-1895 and *B. subtilis* KATMIRA1933 after 24 h incubation. Co-aggregation of probiotic strains with pathogenic bacteria is indicative of competition between the two bacterial species on the attached surfaces, which may play an important role in inhibiting biofilm formation [39].

The minimum biofilm inhibitory concentration is defined as the concentration of an antimicrobial that inhibits either 50% (MBIC-50) or 90% (MBIC-90) of biofilm formation compared to the untreated control group [40]. Our findings were close to the study of Lin et al. [33], who found that MBIC of polymyxin E against *Acinetobacter* spp. isolates was 8.192 µg/mL.

In our study, 7.3–23.3% of the biofilm was prevented when 50% CFS of *B. subtilis* KATMIRA1933 was used. Similarly, no MBIC was determined when *B. amyloliquefaciens* B-1895 CFS was used, while using 50% CFS of *B. amyloliquefaciens* B-1895 prevented 28.4–48.7% of the biofilm formed by the selected *Acinetobacter* spp. isolates (Figure 7).

The antibacterial and anti-biofilm activity of our tested bacilli CFS was reported by Algburi et al. [38] against biofilm formation by *Proteus mirabilis* isolated from urine samples of sheep and patients have urinary tract infections. The authors reported that 25% and 50% CFS of *B. subtilis* KATMIRA1933 prevented 3% and 15.6% of planktonic cells and effectively inhibited 75–84%, respectively, of biofilm formation by *P. mirabilis* of human sources. In regard to *B. amyloliquefaciens* B-1895 CFS, planktonic growth of *P. mirabilis* (human isolate) decreased by 2.9% and 11.3% when the CFS of bacilli at 25% and 50% were used, correspondingly. In addition, 72% and 81% of *P. mirabilis* biofilm (human isolate) were inhibited when 25% and 50% of the CFS were applied, respectively. Similarly, using 25% and 50% of the CFS of *B. amyloliquefaciens* B-1895 caused 38% and 59% biofilm prevention and reduced the density of planktonic cells of *P. mirabilis* (isolated from sheep) by 65.3% and 69.1%, respectively. In the presence of 25% and 50% CFS of *B. subtilis* KATMIRA1933, 51% and 57%, respectively, and the accumulated biofilm *P. mirabilis* isolated from sheep was decreased [38]. The planktonic cells growth and biofilm formation were measured at OD_600_.

The CFS obtained from probiotic species contains various biologically active compounds, including exopolysaccharides, proteins, biosurfactants, and digestive enzymes. These substances are associated with the inhibition or destruction of the preformed biofilm [39].

Our data showed a significant increase in the selected *Acinetobacter* spp. strains growth inhibition when the CFS of bacilli strains was combined with polymyxin E, compared to the use of the antibiotic alone (Figure 8). Other studies have also evaluated antimicrobial combinations of the CFS of probiotic strains with antibiotic discs. For example, Isayenko et al. [10] found an increase in the diameter of the zones of inhibition for *A. baumannii* when antibiotics were combined with the metabolite complexes of *Lactobacillus rhamnosus* (re-classified as *Lacticaseibacillus rhamnosus*) and *Saccharomyces boulardii* using the modified disk-diffusion method. Using the same method, in a recent study, Algburi et al. [41] reported on the complementary activity of cefotaxime combined with the CFS of *B. subtilis* KATMIRA1933 and *B. amyloliquefaciens* B-1895 against methicillin-resistant *Staphylococcus aureus*.

In addition to the disk-diffusion method, a checkerboard assay was also used to evaluate the antimicrobial combinations. Our results showed that the antimicrobial activity of polymyxin E was enhanced when combined with the CFS of tested bacilli against planktonic and biofilm associated cells of the selected *Acinetobacter* spp. isolates. Various studies have been conducted to investigate the nature of antimicrobial reactions. Mathur et al. [42] found synergistic anti-biofilm activity for nisin combined with polymyxins against *P. aeruginosa*: No inhibition in *P. aeruginosa* biofilm was noticed when nisin (1/3× MIC) and polymyxin E (1/5× MIC) were used alone, while significant biofilm prevention was found (more than 15%) when polymyxin E was combined with nisin at the same concentrations. The synergy of polymyxins with other antimicrobials could be an attractive approach for controlling MDR pathogens after ensuring the safety of these combinations on human health.

Probiotic strains produce antimicrobial substances such as organic acids, bacteriocins, hydrogen peroxide, and biosurfactants [43]. The synergistic interactions of biologically active substances produced by probiotics together with antibiotics can (i) increase their antimicrobial activity when used in industrial and medical applications, (ii) reduce the concentrations of both antimicrobials when they are used alone, (iii) and prevent the development of bacterial resistance. These advantages are urgently required to extend the usage of existing antibiotics [44]. Previously, we reported on the synergy between B. subtilis KATMIRA1933-produced subtilosin A and antibiotics (clindamycin and metronidazole) against planktonic cells [45] and biofilms [46] of Gram-variable human pathogen *Gardnerella vaginalis*. Based on the draft genome sequence of *Bacillus amyloliquefaciens* B-1895 [47], the microorganism bears several genes potentially coding for various cyclic peptide antibiotics, some of which may also act synergistically with selected antibiotics and other stressors. Taking probiotics concurrently with antibiotics may reduce the threatening effect of using antibiotics in high concentrations, such as avoiding the risk of developing antibiotic-related dysbiosis [48].

## 4. Materials and Methods

### 4.1. Bacterial Growth Conditions, Isolation, and Identification of Acinetobacter spp. Isolates

In this study, three selected isolates of *Acinetobacter* spp. designated as *Acinetobacter* spp. isolate 1; *Acinetobacter* spp. isolate 2, and *Acinetobacter* spp. isolate 3 were collected from hospitalized patients having burns, wounds, and blood infections, respectively. The samples were initially inoculated on blood agar (HiMedia, Mumbai, India) and MacConkey agar (HiMedia, Mumbai, India). Then, the morphological features of the bacterial colonies were studied. After Gram staining, single suspected colonies (non-lactose fermenting colonies, non-hemolytic, and creamy colonies) were transferred onto MacConkey agar and incubated at 44 °C for 24 h under aerobic conditions to obtain a pure bacterial culture. Bacterial isolates were initially identified using certain biochemical tests, including IMViC tests, catalase, oxidase, urease production, and sugar fermentation in triple sugar iron (TSI) [49]. The identification of the selected *Acinetobacter* spp. isolates was confirmed by a VITEK 2 compact system (BioMerieux, Craponne, France), in which a GN-ID Card contains 64 biochemical tests used to identify Gram-negative bacterial species was manually loaded. According to the manufacturer’s instructions (BioMerieux, Craponne, France), the next steps of bacterial species diagnosis in the VITEK system were performed automatically.

*Bacillus subtilis* KATMIRA1933 and *Bacillus amyloliquefaciens* B-1895 were inoculated into MRS medium (De Man, Rogosa, Sharpe, Becton Dickinson and Company, Sparks, MD, USA) and incubated under aerobic conditions at 37 °C for 24 h.

### 4.2. Ethical Statement and Consent

The samples were collected from patients according to the Institutional Ethical Clearance Committee No.1656 on 22 September 2020. These samples were processed and stored according to the Guiding Principles for Ethical Research issued by the University of Diyala, Baqubah, Iraq.

### 4.3. Antibiogram Assay of the Selected Acinetobacter spp. Isolates and the Probiotic Bacillus Strains

The antibiotic susceptibility of the selected *Acinetobacter* spp. isolates and the tested bacilli strains was evaluated using the Kirby–Bauer method, according to the CLSI guidelines [26]. Briefly, 3–5 colonies of bacterial growth (*Acinetobacter* spp. and *Bacillus* strains) were transferred by a sterile inoculating loop to a tube containing 5 mL of broth culture medium. Using a spectrophotometer (Molecular Diagnostics, Sunnyvale, CA, USA), the bacterial growth was diluted and adjusted to an optical density (OD_630_) of 0.1, which correlated with 10^8^ CFU/mL. Then, 100 μL aliquots of each bacterial suspension were streaked using a swab onto Muller–Hinton agar (MHA) in three directions.

The antibiotics were selected based on the recommendation of physicians as commonly prescribed antimicrobials for *Acinetobacter* infections. The tested antibiotic discs include amikacin (30 μg), colistin (polymyxin E) (25 μg), cefoxitin (30 μg), cefotaxime (30 μg), meropenem (10 μg) and trimethoprim-sulfamethoxazole (1.25/23.75 μg). These antibiotics were placed on the previously inoculated MHA with the selected *Acinetobacter* spp. isolates, and the agar plates were incubated aerobically at 37 °C for 24 h. The diameter of each zone of inhibition was measured in millimeters (mm). Bacterial resistance and sensitivity to antibiotics were determined based on the standard chart approved by the CLSI. Appendix A (CLSI, 2020).

### 4.4. Biofilm Formation Assay

The biofilm formation assay was performed according to Ghellai et al. [50] with minor modifications. Briefly, 20 μL of overnight bacterial growth was diluted into BHI supplemented by 1% glucose (BHIG) to achieve 10^6^ CFU/mL and inoculated into a flat-bottom tissue culture 96-well microplate containing 180 μL of BHIG. A negative control (200 μL of BHIG only) was used in this experiment. The microplate was sealed and incubated at 37 °C for 24 h under aerobic conditions. After incubation, the unattached bacterial cells were removed by pipetting, and the wells were washed twice with PBS (pH 7.1). The microplate was dried at room temperature for 15 min, and the biofilm was fixed by heating for 60 min at 60 °C in the oven. Then, 100 μL of crystal violet solution (0.1%) was added to the treated wells and left for 20 min. After that, the residue of crystal violet was removed, and each well was washed by PBS three times to remove the unbounded crystal violet dye. Then, 200 μL of ethanol 95% was added to the wells, and the plate was then incubated at 4 °C for 30 min to solubilize the crystal violate-stained biofilm mass. The absorbance of the treated and the negative control wells was reported at 630 nm using a microplate reader.

Based on the absorbance, three categories of biofilm formers were identified according to Tang et al. [51]; low biofilm formers (LBF), intermediate biofilm formers (IBF), and high biofilm formers (HBF). When OD * ≤ * ODc indicated non-biofilm, ODc< OD ≤ 2 × ODc = moderately biofilm producer, while 2 × ODc < OD = strong biofilm producer.

* OD: mean optical density of biofilm mass stained with crystal violet; ODc: mean optical density of the negative control.

### 4.5. Preparation of CFS of the Tested Probiotics

The cell-free supernatants (CFS) of *B. subtilis* KATMIRA1933 and *B. amyloliquefaciens* B-1895 were prepared as previously described by Algburi et al. [38]. These strains were inoculated into MRS broth and incubated aerobically at 37 °C for 24–36 h. The bacterial cells were precipitated and removed using a centrifuge (4480× *g* at 4 °C for 30 min). The supernatants were sterilized using a 0.22 μm polytetrafluoroethylene (PTFE) syringe filter (Fisherbrand™, Thermo Fisher Scientific, Waltham, MA, USA). The CFS was kept at 4 °C for less than 5 days before it was used.

### 4.6. Antibiotics Combination with CFS of Probiotics Using Disc Diffusion Method

The antimicrobial combination in the disc was performed using the modified Kirby–Bauer method according to Algburi et al. [41]. Briefly, the overnight growth of the selected *Acinetobacter* spp. isolates in BHI broth was diluted and adjusted to 10^8^ CFU/mL using a spectrophotometer (Molecular Diagnostics, Sunnyvale, CA, USA). Then, the adjusted bacterial growth was streaked over the MHA plate in three directions. Each antibiotic disc was separately saturated with 20 μLof CFS of the tested bacilli. Three types of discs were prepared in this assay: (i) antibiotic disc only, (ii) antibiotic disc saturated with the tested bacilli CFS, and (iii) a blank disc saturated with bacilli CFS only. A blank non-treated disc was used as a negative control. All discs were placed on the surface of MHA, which was previously inoculated with the isolated pathogen. The agar plates were left for 30 min until the antibiotic was diffused from the discs into the surrounding agar surface, and then incubated aerobically at 37 °C for 24 h. After incubation, the bacterial sensitivity and resistance to antimicrobials were identified by measurement of inhibition zones around the discs and according to a standard chart for antibiotic susceptibility testing.

### 4.7. Coaggregation Test

The coaggregation of the tested bacilli with the selected *Acinetobacter* spp. isolates was performed according to Algburi et al. [52] with some modifications. Briefly, the bacterial cultures were harvested from the planktonically grown cells incubated at 37 °C by centrifugation (4480× *g*, 23 °C, 15 min); the cells were then washed with sterile PBS three times. After the third wash, the harvested cells were re-suspended in PBS, and their optical density (OD_630_) was adjusted to 0.25. In a sterile tube, 2 mL of the washed bacilli cells were mixed with 2 mL of the selected *Acinetobacter* spp. growth. As controls, 4 mL of bacterial monoculture was added in separate tubes. The tubes were incubated aerobically at 37 °C, and the OD_630_ values were taken separately at 0, 4, and 24 h. Based on the below equation described by Ledder et al. [53], the coaggregation percentages were calculated.
$$\mathrm{coaggregation}\;\%=\frac{x-y}{x}\times 100$$
where *x* is the OD_630_value before incubation and *y* is the OD_630_value after incubation per-time point.

At 0, 4, and 24 h time points, samples of 5 μL were transferred to a glass slide, stained with Gram staining, and observed for coaggregation score using a transmitted light microscope (Roth GmbH & Co. KG, Karlsruhe, Germany). The bacterial interactions were examined using the 100×/1.25 oil objective. The Kopacam, NIS- Elements D3.0 software was used for photographing. The percentages of coaggregation were analyzed and scored with a system established by Algburi et al. [52], with 0 being the absence of coaggregation and 4 being an abundance of coaggregation. Each experiment was performed in duplicate.

### 4.8. Minimum Inhibitory Concentration (MIC)

MIC determination was performed according to Algburi et al. [54], with minor modifications. Briefly, the 100 mg of polymyxin E was dissolved in 5 mL of sterile distilled water to obtain 20 mg/mL as a stock solution; 1 mL of stock solution was taken and transferred into 19 mL of BHI broth to obtain 1000 µg/mL as the primarily stock solution. A series of two-fold dilutions of antimicrobials (polymyxin E and the CFSs of the tested bacilli) were separately performed with fresh BHI broth into 96-well microplate, with a final volume of 100 µL. Then, 100 μL of the diluted suspension of the selected *Acinetobacter* spp. isolates in BHI (10^6^ CFU/mL) was transferred into each well containing 100 μL of pre-determined concentrations of both antimicrobials. The microplates were incubated under aerobic conditions for 24–36 h at 37 °C. The MICs were determined using a microplate reader (Molecular Diagnostics, Sunnyvale, CA, USA) at OD_630_. The MIC was defined according to the CLSI [26], as the lower concentration of antimicrobial cause bacterial growth inhibition with an OD reading 20% less than the positive control.

### 4.9. Determination of Minimal Biofilm Inhibitory Concentration (MBIC)

The MBIC assay was performed as described in our previous study [54], with minor modifications. Briefly, the stock solution of polymyxin E (1000 µg/mL) was prepared in BHIG broth. The antibiotic and CFS of the tested bacilli were diluted two-fold with fresh BHIG into the 96-well tissue culture microplate broth to a final volume of 100 μL in each well. Separately, the overnight cell culture of the selected *Acinetobacter* spp. isolates was diluted in BHIG broth to 10^7^ CFU/mL. Then, 100 μL as separately added into the wells containing pre-determined concentrations of polymyxin E and bacilli CFS. The microplate was incubated under aerobic conditions for 24–36 h at 37 °C. The non-adherent cells were transferred to a new 96-well microplate, and the absorbance of bacterial growth was evaluated using a microplate reader at OD_630_. The wells were then washed gently twice with 200 μL of PBS. The biofilm was fixed by heating for 60 min at 60 °C and stained with crystal violet, as mentioned above. The absorbance measurement was made using a microplate reader at 630 nm to determine the MBIC.

### 4.10. Checkerboard Assay for Antimicrobial Combinations

To evaluate the antimicrobial potential of the selected bacilli CFSs in combination with polymyxin E against planktonic and biofilm cells of the selected *Acinetobacter* spp. strains, a checkerboard assay was performed following Algburi et al. [46] with minor modifications. Briefly, the 24 h growth of *Acinetobacter* spp. was diluted to achieve 10^6^ CFU/mL. Each antimicrobial agent was diluted two-fold with BHI (to determine MIC) or with BHIG (to determine MBIC) into two separate 96-well microplates. Then, from each dilution of antimicrobial A (CFS of one bacilli strain), 50 μL was taken and added horizontally over 50 μL of each dilution of antimicrobial B (polymyxin E). Then, 100 μL of the selected *Acinetobacter* spp. strains suspension (10^6^ CFU/mL) was separately added to the pre-determined concentrations of antimicrobial combinations. A total of 200 μL of the final bacterial suspension (10^6^ CFU/mL) was added in duplicate, as a positive control. The MIC and MBIC of each antimicrobial combination were determined after 24 h of incubation. After incubation, the non-adherent cells’ growth in the treated wells was evaluated using a microplate reader (Molecular Diagnostics, Sunnyvale, CA, USA) at OD_630_ to determine the MIC of the combinations of the antimicrobials. To determine the MBIC, the wells were gently washed three times with 200 μL of PBS. As previously explained, the biofilm was fixed, stained with crystal violet, and the absorbance was measured at 630 nm using a microplate reader (Molecular Diagnostics, Sunnyvale, CA, USA) according to Algburi et al. [54]; isobolograms were used to analyze the nature of antimicrobial combinations or synergistic, antagonistic, or additive activity against the planktonic cells. The total fractional inhibitory concentrations index (ΣFIC) was used to evaluate the anti-biofilm potential of antimicrobial combinations against the selected *Acinetobacter* spp. isolates.

### 4.11. Checkerboard Assay, Data Analysis

Isobolograms were used to compare the MIC values of each antimicrobial agent alone with its MIC values in combinations with other antimicrobial agents. Data were analyzed and explained as previously described by Turovskiy and Chikindas [55]. The ΣFIC was calculated using the following equations:ΣFIC = FICA + FICB
FICA = (CA/MICA), FICB = (CB/MICB)
where MICA and MICB are the MICs of antimicrobials A and B alone, respectively, and CA and CB are the concentrations of the antimicrobials in combination.

An FIC index of <0.5 indicates synergism, >0.5 to <1 indicates additive effects, >1 to <2 indifference, and ≥2 antagonism [56].

### 4.12. Statistical Analysis

The data obtained in this study were sorted according to the graph pad prism V5 software. In this study, the two-way ANOVA test and Chi-square test were performed to analyze the effect of bacilli CFS and polymyxin E on bacterial growth. *p* values of <0.05 were considered statistically significant. Sigma plot V11 software was used to generate the isobolograms for the antimicrobial combinations against the planktonic cells of the isolated *Acinetobacter* spp.

## 5. Conclusions

*Acinetobacter* spp., one of the most important pathogens of several hospital-acquired infections, showed high resistance to most tested antibiotics but was highly sensitive to polymyxin E. Our findings should attract researchers to implement a strict protocol to control such infections caused by XDR and MDR isolates of selected *Acinetobacter* spp. Strong biofilm formation by the majority of *Acinetobacter* spp. isolates is improving their colonization and antibiotic resistance. Probiotics and the natural antimicrobials they produce are good candidates for use as alternative agents for controlling biofilm-associated *Acinetobacter* spp. stains. Our data provide insight into the development of novel, safe, and effective antimicrobial and anti-biofilm agents to prevent biofilm-associated multidrug-resistant infections. It was reported that natural compounds produced by beneficial microbes can target bacterial cell envelopes, and as a result, they may facilitate antibiotics’ activity and reduce the possibility of antibiotics resistance. The antimicrobial and anti-biofilm activities of polymyxin E were improved and showed synergism when combined with the CFS of the tested probiotic bacilli against planktonic and biofilm-associated cells of the selected *Acinetobacter* spp. isolates. Future in vivo studies are needed to clarify the mechanism (pharmacokinetic and pharmacodynamic) of polymyxins when combined with substances produced by probiotics to inhibit the biofilm of *Acinetobacter* spp. isolates and to ensure the safety of these antimicrobial interactions.

## Figures and Tables

**Figure 1 pathogens-10-01574-f001:**
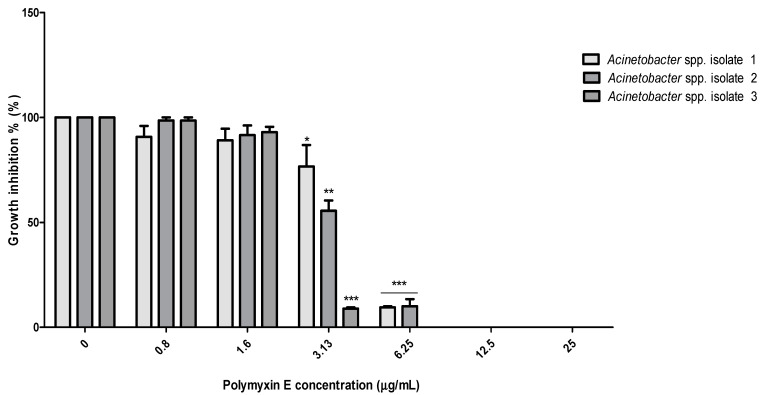
Antibacterial activity of polymyxin E against the selected *Acinetobacter* spp. isolates 1, 2, and 3. Data represented as mean MIC ± SEM (µg/mL) to three independent experiments. Asterisks refer to significance levels: * *p* < 0.05, ** *p* < 0.01, and *** *p* < 0.001.

**Figure 2 pathogens-10-01574-f002:**
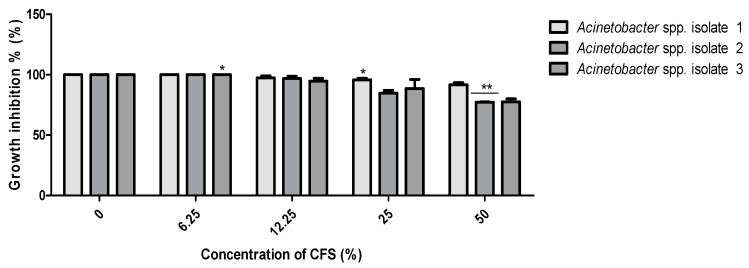
Antibacterial activity of *B. subtilis* KATMIRA1933 CFS against the selected *Acinetobacter* spp. isolates 1, 2, and 3. Data presented as mean MIC ± SEM (µg/mL) to three independent experiments. Asterisks refer to significance levels: * *p* < 0.05, and ** *p* < 0.01.

**Figure 3 pathogens-10-01574-f003:**
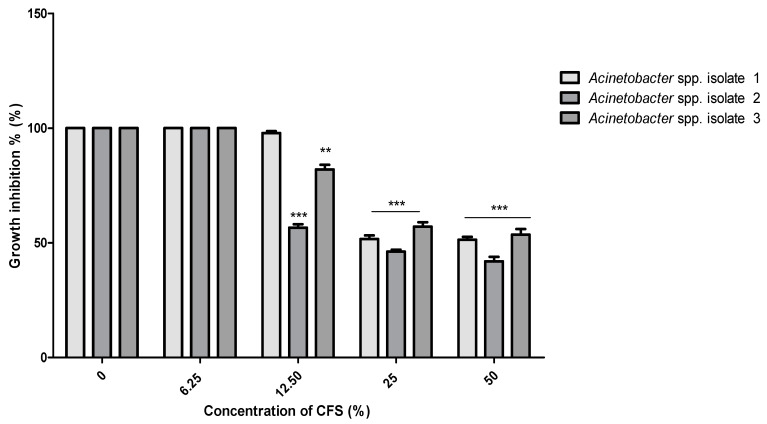
Antibacterial activity of *B. amyloliquefaciens* B--1895 CFS against the selected *Acinetobacter* spp. isolates 1, 2, and 3. Data presented as mean MIC ± SEM (µg/mL) to three independent experiments. Asterisks refer to significance levels: ** *p* < 0.01, and *** *p* < 0.001.

**Figure 4 pathogens-10-01574-f004:**
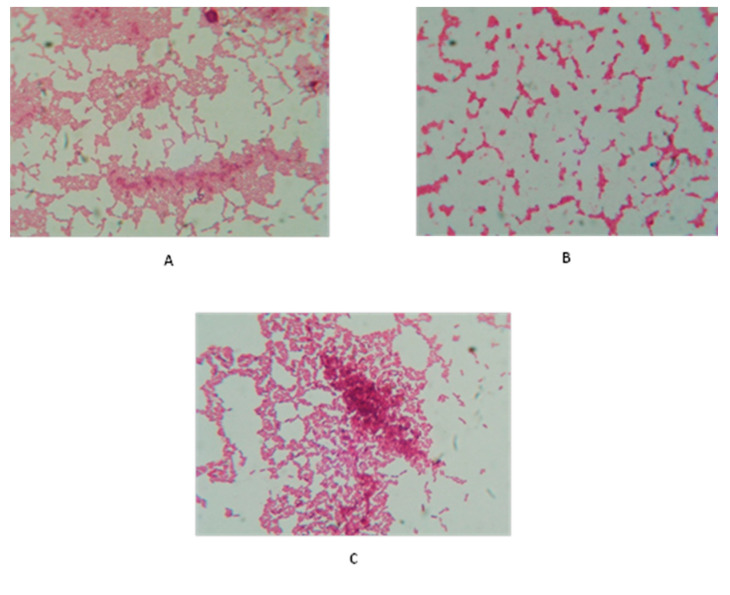
Auto and coaggregation of the tested bacilli strains with the selected *Acinetobacter* spp. isolates. (**A**) Autoaggregation of *B. amyloliquefaciens* B-1895; (**B**) autoaggregation of *Acinetobacter* spp. isolates (some auto-aggregation); (**C**) coaggregation of *B. amyloliquefaciens* B-1895 with *Acinetobacter* spp. isolates (coaggregation). Bacterial auto- and coaggregations were captured under a biological microscope using oil immersion at 1000× magnification; the scale bar is 10 µL.

**Figure 5 pathogens-10-01574-f005:**
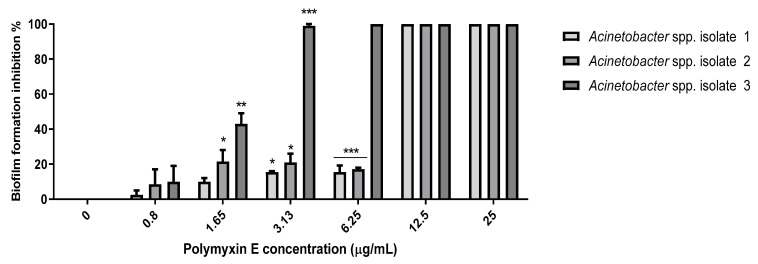
Anti-biofilm activity of polymyxin E against the selected *Acinetobacter* spp. isolates 1, 2, and 3. Data presented as mean MIC ± SEM (µg/mL) to three independent experiments. Asterisks refer to significance levels: * *p* < 0.05, ** *p* < 0.01, and *** *p* < 0.001.

**Figure 6 pathogens-10-01574-f006:**
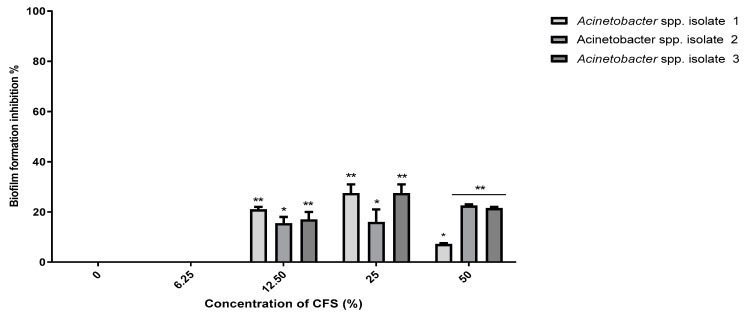
Anti-biofilm activity of *B. subtilis* KATMIRA1933 CFS against the selected *Acinetobacter* spp. isolates 1, 2, and 3. Data presented as mean MIC ± SEM (%) to three independent experiments. Asterisks refer to significance levels: * *p* < 0.05, and ** *p* < 0.01.

**Figure 7 pathogens-10-01574-f007:**
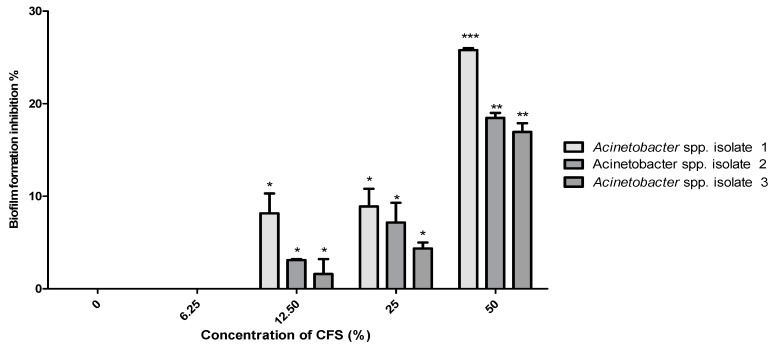
Anti-biofilm activity of *B. amyloliquefaciens* B-1895 CFS against the selected *Acinetobacter* spp. isolates 1, 2, and 3.. Data presented as mean MIC±SEM (%) to three independent experiments. Asterisks refer to significance levels: * *p* < 0.05, ** *p* < 0.01 and *** *p* < 0.001.

**Figure 8 pathogens-10-01574-f008:**
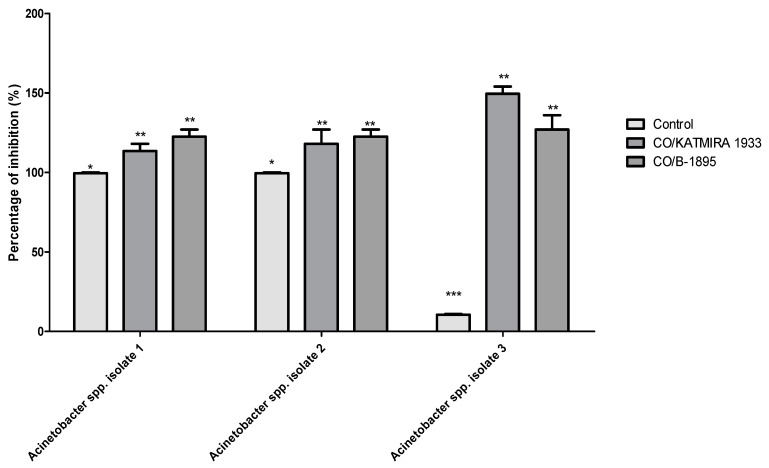
Zone of the selected *Acinetobacter* spp. isolates growth inhibition when polymyxin E was used alone and in combination with the CFS of the tested probiotic strains using disc diffusion method. Data presented as mean ± SEM to three independent experiments. Asterisks refer to significance levels: * *p* < 0.05, ** *p* < 0.01, and *** *p* < 0.001.

**Figure 9 pathogens-10-01574-f009:**
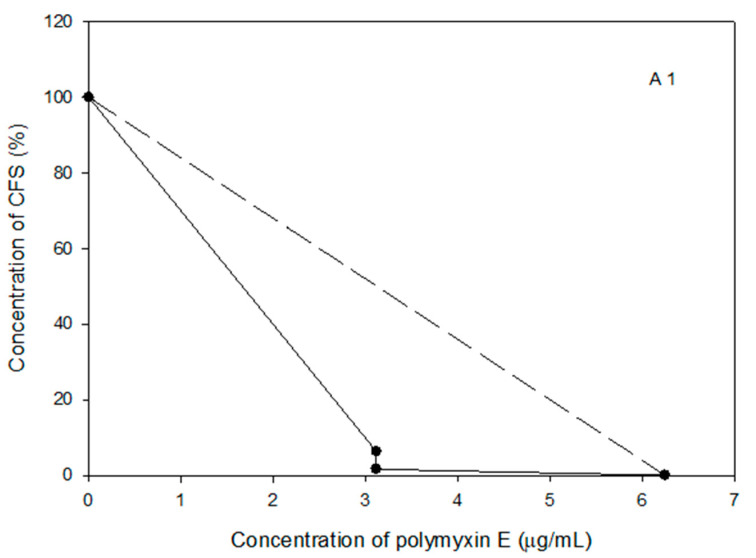
(**A**,**B**) (**1**–**3**). Isobolograms of polymyxin E in combination with CFS of probiotic strains against planktonic cells of the selected *Acinetobacter* spp. (**A**) *B. amyloliquefaciens* B-1895, (**B**) *B. subtilis* KATMIRA1933), (**1**–**3**) referred to the *Acinetobacter* spp. isolate numbers. The dashed lines connect the two MIC values (closed circles), while the solid lines connect the sub-MIC vlues (closed circles below the dashed lines).

**Table 1 pathogens-10-01574-t001:** Resistance rate of selected *Acinetobacter* spp. isolates to the selected antibiotics.

Antibiotics Family	Antibiotics Type	Resistant Isolates No. and %
Penicillins	Ampicillin	*Acinetobacter* spp. isolates 1, 2, and 3
B-lactam inhibitors	Piperacillin/Tazobactam	*Acinetobacter* spp. isolates 1, 2, and 3
Cephalosporins	Cefazolin	*Acinetobacter* spp. isolates 1, 2, and 3
	Cefoxitin	*Acinetobacter* spp. isolates 1, 2, and 3
	Ceftazidime	*Acinetobacter* spp. isolates 1, 2, and 3
	Ceftriaxone	*Acinetobacter* spp. isolates 1, 2, and 3
	Cefepime	*Acinetobacter* spp. isolates 1, 2, and 3
Carbapenems	Imipenem	*Acinetobacter* spp. isolates 1, 2, and 3
	Meropenem	*Acinetobacter* spp. isolates 1, 2, and 3
Aminoglycosides	Amikacin	*Acinetobacter* spp. isolates 1 and 2
	Gentamicin	*Acinetobacter* spp. isolates 1 and 2
	Tobramycin	*Acinetobacter* spp. isolates 1 and 2
Tetracyclines	Tigecycline	*Acinetobacter* spp. isolate 1
	Minocycline	*Acinetobacter* spp. isolate 1
Fluoroquinolones	Ciprofloxacin	*Acinetobacter* spp. isolates 1, 2, and 3
	Levofloxacin	*Acinetobacter* spp. isolates 1, 2, and 3
Folate pathway antagonists	Trimethoprim /sulfamethoxazole	*Acinetobacter* spp. isolates 1 and 2
Lipopeptides	Polymyxin E	None

**Table 2 pathogens-10-01574-t002:** Antibiotic susceptibility of probiotic strains.

Probiotic Strains	The Average Diameters of Inhibition Zone around Antibiotic Discs (mm)					
	AK *	PME *	FOX *	CTX *	MEM *	TS *
*Bacillus subtilis* KATMIRA1933	22	zero	35	40	37	37
*Bacillus amyloliquefaciens* B-1895	14	zero	28	28	35	30

* (AK) amikacin 30 mg, (PME) polymyxin E 25 mg, (FOX) cefoxitin 30 mg, (CTX) cefotaxime 30 mg, (MEM) meropenem 10 mg, (TS) trimethoprim-sulfamethoxazole 1.25/23.75 mg.

**Table 3 pathogens-10-01574-t003:** Auto- and coaggregation of the tested bacilli strains at 4 h and 24 h of incubation.

Bacterial Strains	Auto- and Co-Aggregation % after 4 h	Auto- and Co-Aggregation % after 24 h
*B.**amyloliquefaciens* B-1895	25.5%	95.7%
*B. subtilis* KATMIRA1933	22.76%	82.4%
*Acinetobacter* spp. isolates1, 2, and 3	(24.5%, 20.58%, 28.97%)	(0.48%, 8.4%, 30.5%)
*B.**amyloliquefaciens* B-1895 + *Acinetobacter* spp. isolate 1	23.98%	50.57%,
*B.**amyloliquefaciens* B-1895 + *Acinetobacter* spp. isolate 2	29.39%	55.64%
*B.**amyloliquefaciens* B-1895 + *Acinetobacter* spp. isolate 3	17.15%	50.67%
*B. subtilis* KATMIRA1933 + *Acinetobacter* spp. isolate 1	33.43%	60.1%,
*B. subtilis* KATMIRA1933 + *Acinetobacter* spp. isolate 2	31.89%	53.16%
*B. subtilis* KATMIRA1933 + *Acinetobacter* spp. isolate 3	34%	62.8%

## Data Availability

Not applicable.

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
