# Peer review of "Antimicrobial and Anti-Biofilm Activity of Polymyxin E Alone and in Combination with Probiotic Strains of Bacillus subtilis KATMIRA1933 and Bacillus amyloliquefaciens B-1895 against Clinical Isolates of Selected Acinetobacter spp.: A Preliminary Study"

_pathogens, 2021, doi:10.3390/pathogens10121574_

Round 1
Reviewer 1 Report
I again thank the authors for their response. However, data to address the mechanistic basis for the enhanced activity or the active component in CFS was still not provided. Interestingly, the authors have a promising lead in subtilosin A. If new data is provided that tests purified subtilosin A in the experiments, this would provide important new information.
Author Response
Dear respected Reviewer, please see our response below to your kind comment:
The authors are grateful to the Reviewer for very valuable comments and suggestions on the improvement of the manuscript and its appeal to the reader. All of the Reviewer’s comments were addressed in the revised version of the manuscript. Specifically, the manuscript’s title was changed to emphasize the preliminary nature of the report and appropriate changes, based on the Reviewer’s suggestion, were introduced in the manuscript’s Discussion chapter and its Conclusion, highlighting limitations of the report and the need for mechanistic studies.
The authors fully agree with the Reviewer: mechanistic studies should be conducted and there is good preliminary evidence allowing the authors and the Reviewer to speculate on the possible role of subtilosin A. However, as the authors mentioned previously, the report on mechanistic studies will require a complete restructuring of the manuscript, with a re-direction of its major strategy from the activity against clinical isolates to fundamental molecular mechanisms of the observed phenomenon. The authors feel inspired by the Reviewer to conduct these studies. Right now, the authors feel the inclusion of mechanistic investigation will deviate the reader from the major take-home message of this report.
Reviewer 2 Report
pathogens-1472672-peer-review-v1_08 November 2021
The present work is an example of link between academic study and real potential application on combination between known antimicrobial (polymixin) and probiotic strains with antimicrobial properties. Authors have covered appropriate research plan in order to determine and show potential synergetic interactions between studied antimicrobials. In my opinion,some corrections and adjustments will need to be taken into considerations.
Ln17, 21, and other places in the manuscript, correct to "spp."
Maybe on figure 4 dimension bares can be added?
I preferer to call selected Acinetobacter ssp. strain, since some of the performed test showing that this is different isolates, so, can be considered as unique strains.
Author Response
- The present work is an example of link between academic study and real potential application on combination between known antimicrobial (polymixin) and probiotic strains with antimicrobial properties. Authors have covered appropriate research plan in order to determine and show potential synergetic interactions between studied antimicrobials. In my opinion, some corrections and adjustments will need to be taken into considerations.
Response:
The authors are grateful to the Reviewer for the effort in evaluating the submitted manuscript.
- Ln17, 21, and other places in the manuscript, correct to "spp."
Response:
“spp” was corrected to “spp.” in the whole manuscript
- Maybe on figure 4 dimension bares can be added?
Response:
Figure 4 was implemented to clarify the strength of bacterial interactions (auto and coaggregations) using the light microscope magnification 100X with a scale bar 10 µL. The following statement was added to the figure 4 legend “Bacterial auto and coaggregations were captured under a biological microscope using oil immersion at 1000x magnification, the scale bar is 10 µL”
- I preferer to call selected Acinetobacter ssp. strain, since some of the performed test showing that this is different isolates, so, can be considered as unique strains.
Response:
The authors appreciate the kind opinion of the Reviewer. As you suggest, it was called “selected Acinetobacter spp. strain” where we think it is necessary. (please see the revised manuscript).
Reviewer 3 Report
This study aimed to evaluate the antibacterial and anti-biofilm activities of the tested probiotic bacilli and their cell-free supernatants (CFS) alone and in combination with polymyxin E against Acinetobacter spp isolates. The manuscript is well written. However, there are some minor comments to address
1- Please provide the ethical statement in a separate section
2- Be sure that the name of the bacteria is italic throughout the manuscript
3- Line 571: add space: then incubated
4- The appendix table can be provided as a supplementary table
5- Did you confirm your isolated colonies using PCR?
6- Do you mean by tolerant isolates that they were sensitive to the tested antibiotic.
7- Line 97-107: you can only refer to table 1
8- Please present your data in figure 1, 2, 3 as growth inhibition %
9- Please revise figure 4 legend and make it more clear
10- Figure 5 and 6: please change to biofilm formation inhibition
11- Figure 8: calculate the % of inhibition for each isolate
12- Figures 9, 10, 11: please combine in one figure and put all of these figures together
Author Response
This study aimed to evaluate the antibacterial and anti-biofilm activities of the tested probiotic bacilli and their cell-free supernatants (CFS) alone and in combination with polymyxin E against Acinetobacter spp isolates. The manuscript is well written. However, there are some minor comments to address.
Response:
The authors would like to appreciate the Reviewer’s effort in evaluating our submitted manuscript.
- Please provide the ethical statement in a separate section
Response:
The authors provided an” Ethical Statement and Consent” in a separate section
- Be sure that the name of the bacteria is italic throughout the manuscript.
Response:
The names of the bacteria were checked throughout the manuscript and changed to italic form.
- Line 571: add space: then incubated
Response:
The space is added. Please see Ln#586
- The appendix table can be provided as a supplementary table
Response:
The authors considered the appendix table a supplementary.
- Did you confirm your isolated colonies using PCR?
Response:
Since it is a preliminary study, the authors used VITEK 2 compact system to identify the bacterial isolates based on their biochemical reactions (activity). In this study, the isolated pathogens were indicated as “selected Acinetobacter spp. isolates”. The future mechanistic study of the newly isolated pathogens will include their taxonomic identification (an analysis of the rpoB gene) using the appropriate approach, such as using PCR.
- Do you mean by tolerant isolates that they were sensitive to the tested antibiotic.
Response:
The authors mean by “tolerant isolates” that they were “resistant” to the tested antibiotics.
- Line 97-107: you can only refer to table 1
Response:
The authors referred to table 1, only, at the end of the paragraph.
- Please present your data in figure 1, 2, 3 as growth inhibition %
Response:
Our data in figure 1, 2, 3 were presented as growth inhibition % and changed accordingly in the text of the manuscript, as suggested by the Reviewer.
- Please revise figure 4 legend and make it more clear
Response:
The figure 4 legend was revised and the typo was deleted. Now, the legend is clearer.
- Figure 5 and 6: please change to biofilm formation inhibition
Response:
Our data in figure 5 and 6 were presented as biofilm formation inhibition %, and changed accordingly in the text of the manuscript,as suggested by the Reviewer.
- Figure 8: calculate the % of inhibition for each isolate
Response:
The % of biofilm inhibition was calculated for each isolate.
- Figures 9, 10, 11: please combine in one figure and put all of these figures together
Response:
The authors have combined figures 9, 10, 11 in one figure and put all of them together under one legend titled “Figure 9 A & B (1-3). Isobolograms of polymyxin E in combination with CFS of bacilli strains against planktonic cells of the selected Acinetobacter spp. (A; B. amyloliquefaciens B-1895, B; B. subtilis KATMIRA1933), (1, 2, 3) referred to the Acinetobacter spp. isolate numbers”.
Round 2
Reviewer 3 Report
Thank you for addressing all the comments
This manuscript is a resubmission of an earlier submission. The following is a list of the peer review reports and author responses from that submission.
Round 1
Reviewer 1 Report
Acinetobacter baumannii, is one of the most important pathogens of several hospital-acquired infections showed high resistance to most tested antibiotics but was highly sensitive to colistin. Their findings should attract researchers to implement a strict protocol to control such infections caused by XDR and MDR isolates of A. baumannii. The authors showed a strong biofilm formation by the majority of A. baumannii and suggested that it is improving their colonization and antibiotic resistance. They also suggested that probiotics and their natural antimicrobials are good candidates used as alternative agents for controlling biofilm-associated A. baumannii.
There are minor changes that I suggest, line 288, 316, 369, and 400 in the Discussion, "Table 1, figures 2 and 7" should be deleted. I understand that it points readers to the correct tables and figures, however, it is not needed, unless if the Journal require that it be done like that.
Line
Author Response
Reviewer #1
Comments and Suggestions for Authors
Acinetobacter baumannii, is one of the most important pathogens of several hospital-acquired infections showed high resistance to most tested antibiotics but was highly sensitive to colistin. Their findings should attract researchers to implement a strict protocol to control such infections caused by XDR and MDR isolates of A. baumannii. The authors showed a strong biofilm formation by the majority of A. baumannii and suggested that it is improving their colonization and antibiotic resistance. They also suggested that probiotics and their natural antimicrobials are good candidates used as alternative agents for controlling biofilm-associated A. baumannii.
Response:
The authors are grateful to the Reviewer for the effort on evaluation of the submitted manuscript.
There are minor changes that I suggest, lines 288, 316, 369, and 400 in the Discussion, "Table 1, figures 2 and 7" should be deleted. I understand that it points readers to the correct tables and figures, however, it is not needed, unless if the Journal requires that it be done like that.
Response:
The authors understand the Reviewer’s suggestion to make the manuscript less wordy; however, as it was mentioned by the Reviewer, perhaps, this should be left for the Handling Editor to decide.
Reviewer 2 Report
This manuscript describes the effects of polymyxin E (colistin) alone and in combination with cell free supernatants from two probiotic Bacillus spp on A. baumannii. Various parameters were tested including (i) antibacterial activity alone and in combination with cell free supernatants, (ii) aggregation of A. baumannii with probiotic strains and (iii) anti-biofilm activity of CFS. The manuscript has deficiencies that need to be addressed.
- A major weakness of the study is that there is no data that addresses either the mechanistic basis for the enhanced activity of colistin + CFS or the active component in CFS responsible for the enhancement.
- The identification system used to identify strains as A. baumannii is unable to discriminate between A. baumannii and A. nosocomialis, which requires an analysis of the rpoB gene. Therefore, reporting these strains as A. baumannii may be incorrect. Why were well defined strains not used?
- In Fig. 2, what are the p values referring to? Is it between the strain(s) at 50 % or between a strain at 50% and 0% CFS? The same issue applies to Figs. 3, 5, 6, 7, and 8.
- What is the significance of the aggregation and co-aggregation studies in Table 3 and Fig. 4? In Fig. 4c, how was each cell type determined?
- In figure 5, the effect of colistin on biofilm formation are essentially identical as the effects on cell growth in Fig. 1. This likely explains the lack of biofilm, the cells simply did not grow.
- Line 97 and throughout the manuscript: Resistant should be used in place of tolerant.
Author Response
Reviewer #2
Comments and Suggestions for Authors
This manuscript describes the effects of polymyxin E (colistin) alone and in combination with cell-free supernatants from two probiotic Bacillus spp on A. baumannii. Various parameters were tested including (i) antibacterial activity alone and in combination with cell-free supernatants, (ii) aggregation of A. baumannii with probiotic strains and (iii) anti-biofilm activity of CFS. The manuscript has deficiencies that need to be addressed.
Response:
The authors appreciate the Reviewer’s effort on the thorough critical analysis of the submitted manuscript.
- A major weakness of the study is that there is no data that addresses either the mechanistic basis for the enhanced activity of colistin + CFS or the active component in CFS responsible for the enhancement.
Response:
The authors agree with this statement of the Reviewer. In fact, the goal of the study was a preliminary evaluation of a possible effect of probiotics-derived bioactives on clinical isolates, with emphasis placed on freshly isolated pathogens, which are usually more resistant to various stresses (including antimicrobial agents) as compared to laboratory strains.
- The identification system used to identify strains as A. baumannii is unable to discriminate between A. baumannii and A. nosocomialis, which requires an analysis of the rpoB gene. Therefore, reporting these strains as A. baumannii may be incorrect. Why were well-defined strains not used?
Response:
The authors agree with this comment. For this effort, the authors were focused on their primary objective of finding if the probiotics-produced bioactives can assist in controlling biofilms of freshly isolated Acinetobacter. Indeed, the Reviewer is right when mentioning that one must be particularly careful when assigning a newly isolated Acinetobacter to one or another species. For instance, the inaccuracies in taxonomical identifications were shown by those using MALDI-TOF MS approach, which became valid only with modification of the method (doi: 10.1016/j.syapm.2013.08.001). Certainly, as the Reviewer pointed, rpoB gene comparative analysis will provide the most accurate result (doi: 10.1371/journal.pone.0104882). This most valuable comment is well-taken. The authors changed the manuscript wording, saying “Acinetobacter sp.”, and the Discussion session of the manuscript will contain an appropriate statement, indicating that the mechanistic study on the newly isolated pathogens will include their taxonomic identification using an appropriate approach. The authors prefer to leave this for a separate report on mechanistic studies.
- In Fig. 2, what are the p values referring to? Is it between the strain(s) at 50 % or between a strain at 50% and 0% CFS? The same issue applies to Figs. 3, 5, 6, 7, and 8.
Response:
In Fig. 2 and other figures, the p-value refers to the differences between the strains at 0% CFS concentration (control) and other concentrations.
- What is the significance of the aggregation and co-aggregation studies in Table 3 and Fig. 4? In Fig. 4c, how was each cell type determined?
Response:
Aggregation of probiotic is a preliminary step (or test) to evaluate their capability for biofilm formation. Co-aggregation of probiotic strains with pathogenic bacteria is indicative of competition between the two bacterial species on the attached surfaces, which may play an important role in inhibiting biofilm formation by pathogenic bacteria. In Fig 4c, each cell type was determined based on their reaction to Gram-stain, Bacillus spp are Gram-positive bacilli, while Acinetobacter spp are Gram-negative bacilli.
- In figure 5, the effect of colistin on biofilm formation are essentially identical as the effects on cell growth in Fig. 1. This likely explains the lack of biofilm, the cells simply did not grow.
Response: The authors would like to thank the Reviewer for this observation. By mistake, the y-axis of Fig. 5 was not properly identified in the original version of the manuscript. Figure 5 is not referring to the effect of colistin on cell growth but to the biofilm mass (%) formed by Acinetobacter spp. Fig. illustrates the effects of colistin on the bacterial cell growth
- Line 97 and throughout the manuscript: Resistant should be used in place of tolerant.
Response:
The authors use the term “tolerant” as looser and more phenotypical/observational as compared to “resistant” which implies mutation.
Reviewer 3 Report
pathogens-1414948-peer-review-v1
The present work is an example of link between academic study and real potential application on combination between known antimicrobial (polymixin) and probiotic strains with antimicrobial properties. Authors have covered appropriate research plan in order to determine and show potential synergetic interactions between studied antimicrobials. In my opinion , several corrections and adjustments will need to be taken into considerations.
L21: Since the A. baumannii cultures were differentiated and identified, then will be better to call them "strains". Please, correct to: Three strains of A. baumannii....
Please, in results/discussion sections, author’s needs to provide some comments in supporting that 3 selected A. baumannii cultures are really different strains. Well, some of the performed initial test, including antibiotic resistance was showing different behavior, however, authors needs to give additional focus and provide arguments that this are different strains and not replica of the same strain isolated in multiple occasions, even isolated from different patience.
Ln35: Please, add italics for "baumannii"
Ln40: Please, abbreviate Acinetobacter
Ln40: Please, check, this needs to be Enterococcus faecium or Enterococcus faecalis?
Ln75-76. This last sentence of the introduction is kind of repetition of the previous two sentence. Maybe can be deleted. Please, consider changing.
Please, in the text, if you have sufficient evidences that 3 A. baumannii cultures are different, will be more appropriate to call them strains and not isolates.
In Table 1, instead to state "All", maybe will be better to list the name of the 3 cultures.
Please, explain BSK, presume that this is Bacillus subtilis Katamira?
Figure 1 was presented as bacterial growth; Figure 2 as Percentage of bacterial growth and Figure 3 as Bacterial survival. I am suggesting that authors can present these 3 figures in same way, same style of their Y axe in order to facilitate reading and comprising of the presented data.
As well, on X axe, please, present symbols in same way.
Moreover, in legend of Fig 3, explain BAB. Please, use same style for abbreviations: Maybe BSK-1933 (on Fig 2, you have stated as BSK 1933).
For figure 2 and 3, on X axe word concentration is appropriate?
Ln163: Please, add interval between B. and amyloliquefaciens
Table 3: Please, add interval before %
Table 3: Maybe 3 strains of A. baumannii need to be presented on 3 separated lines, not combined. Will be easier to follow the shown results.
Legend of Figure 4: Please, add italics for the bacterial names and pay attention on the intervals between words
Please, use word polymyxin E or colistin, but do not change between both in the text. Please, see Ln174 and legend of fig 5. In other places of the manuscript, as well can be observed similar cases, then needs to be corrected.
Fig. 6 and 7. Is words "biofilm mass %" the best way to describe Y axe? Maybe can be used "biofilm integrity"?
Ln337: Please, add interval after (colistin).
Ln396-399: Please, check carefully for intervals between words. Please, scan enters manuscript for similar typos.
Ln453: Is word collected really appropriate in this context?
Ln482: Maybe statement "The antibiotics were selected based on the recommendation of the physicians" needs to be replaced with recommendations from previous work, or as recommended by specific Health agency?
Ln516: Filters were provided from what supplier? Why 0.45, and not 0.22?
Ln558: Maybe replace 20000 μg/mL with 20 mg/mL?
References needs to be check manually with attention to intervals between words, use of capitals, page numbers of cited works, abbreviations of the journal, use of bold.
Author Response
Reviewer #3
Comments and Suggestions for Authors
pathogens-1414948-peer-review-v1
The present work is an example of the link between academic study and real potential application on combination between known antimicrobial (polymixin) and probiotic strains with antimicrobial properties. Authors have covered appropriate research plan in order to determine and show potential synergetic interactions between studied antimicrobials. In my opinion , several corrections and adjustments will need to be taken into considerations.
Response:
The authors appreciate the Reviewer’s critical analysis of their report aimed at improvement of the manuscript and its appeal to the reader.
L21: Since the A. baumannii cultures were differentiated and identified, then will be better to call them "strains". Please, correct to: Three strains of A. baumannii....
Response:
The isolated pathogenic microorganisms will be referred to as strains, once their identity is clarified with the assistance of appropriate methods, during the already planned mechanistic studies (please, see the authors’ response to Question #2 from Reviewer #2). Until then, the studied microorganisms will be referred to as “Acinetobacter spp isolates”.
Please, in results/discussion sections, author’s needs to provide some comments in supporting that 3 selected A. baumannii cultures are really different strains. Well, some of the performed initial test, including antibiotic resistance was showing different behavior, however, authors needs to give additional focus and provide arguments that this are different strains and not replica of the same strain isolated in multiple occasions, even isolated from different patience.
Response:
Please, see the authors’ response to the Reviewer’s previous comment.
Ln35: Please, add italics for "baumannii"
Response:
As per Reviewer #2 Comment #2, these are all changed to “Acinetobacter spp isolates”.
Ln40: Please, abbreviate Acinetobacter
Response:
Since the species-level identification will be performed at the stage of the mechanistic study, the investigated microorganism will be referred to as “Acinetobacter spp isolates”.
Ln40: Please, check, this needs to be Enterococcus faecium or Enterococcus faecalis?
Response:
It was corrected to Enterococcus faecalis, thanks for your kind notice.
Ln75-76. This last sentence of the introduction is kind of repetition of the previous two sentence. Maybe can be deleted. Please, consider changing.
Response:
As per the Reviewer’s suggestion, this sentence was deleted.
Please, in the text, if you have sufficient evidences that 3 A. baumannii cultures are different, will be more appropriate to call them strains and not isolates.
Response:
Please, see the authors’ response to the Reviewer’s previous comments.
In Table 1, instead to state "All", maybe will be better to list the name of the 3 cultures.
Response:
The Reviewer’s comment is very important. Indeed, since we are talking about isolates, these must be named “Acinetobacter spp isolate 1; Acinetobacter spp isolate 2, and Acinetobacter spp isolate 3”.
Please, explain BSK, presume that this is Bacillus subtilis Katamira?
Response:
The Reviewer is correct: BSK refers to Bacillus subtilis KATMIRA1933 and it was changed accordingly.
Figure 1 was presented as bacterial growth; Figure 2 as Percentage of bacterial growth and Figure 3 as Bacterial survival. I am suggesting that authors can present these 3 figures in same way, same style of their Y axe in order to facilitate reading and comprising of the presented data.
Response:
We presented the 3 figures in the same way, the same style of their Y axes.
As well, on X axe, please, present symbols in same way.
Response:
The X-axes are presented with symbols in the same manner.
Moreover, in legend of Fig 3, explain BAB. Please, use same style for abbreviations: Maybe BSK-1933 (on Fig 2, you have stated as BSK 1933).
Response:
Both BAB and BSK were explained in the legend and inside the figures, now changed as suggested by the Reviewer.
For figure 2 and 3, on X axe word concentration is appropriate?
Response:
Word “concentration” was replaced and explained accordingly.
Ln163: Please, add interval between B. and amyloliquefaciens
Response: done.
Table 3: Please, add interval before %
Response: done.
Table 3: Maybe 3 strains of A. baumannii need to be presented on 3 separated lines, not combined. Will be easier to follow the shown results.
Response: done
Legend of Figure 4: Please, add italics for the bacterial names and pay attention on the intervals between words
Response: done.
Please, use word polymyxin E or colistin, but do not change between both in the text. Please, see Ln174 and legend of fig 5. In other places of the manuscript, as well can be observed similar cases, then needs to be corrected.
Response:
Polymyxin E and colistin are often used interchangeably. The authors acknowledge the request as making sense to avoid the reader’s confusion. The manuscript was modified for uniformity in the name of the drug. Antibiotic’s name “polymyxin E” was used instead of “colistin”
Fig. 6 and 7. Is words "biofilm mass %" the best way to describe Y axe? Maybe can be used "biofilm integrity"?
Response:
This is indeed a % of the biofilm mass, which is to some point is an indication of the biofilm integrity.
Ln337: Please, add interval after (colistin).
Response: done (Ln 489 original submission).
Ln396-399: Please, check carefully for intervals between words. Please, scan enters manuscript for similar typos.
Response: this is a commonly observed flaw caused by MS Word program when the files are transferred between multiple users (such as co-authors). The authors will make everything possible to avoid/fix this matter.
Ln453: Is word collected really appropriate in this context?
Response:
The word “obtained” is more appropriate (changed, Ln 621 in the original submission).
Ln482: Maybe statement "The antibiotics were selected based on the recommendation of the physicians" needs to be replaced with recommendations from previous work, or as recommended by specific Health agency?
Response (Ln 486, original submission):
This is a recommendation for treatment at local hospital which follows the commonly accepted scenarios such as described here: DOI: 10.1517/14656561003596350.
Ln516: Filters were provided from what supplier? Why 0.45, and not 0.22?
Response:
This was a typo (original submission Ln 520-521), replaced with: “The supernatants were sterilized using a 0.22 μm polytetrafluoroethylene (PTFE) syringe filter (Fisherbrand™, Thermo Fisher Scientific, Waltham, MA, USA).”
Ln558: Maybe replace 20000 μg/mL with 20 mg/mL?
Response:
Done (Ln 562 of the original submission).
References needs to be check manually with attention to intervals between words, use of capitals, page numbers of cited works, abbreviations of the journal, use of bold.
Response:
The authors manually checked the list of references according to the journal’s requirements.
Round 2
Reviewer 2 Report
I thank the authors for the corrections. However, it is my view that this study is still too preliminary without data that addresses either the mechanistic basis for the enhanced activity of colistin + CFS or the active component in CFS responsible for the enhancement.